# Comparison of the Safety and Efficacy between Preserved and Preservative-Free Latanoprost and Preservative-Free Tafluprost

**DOI:** 10.3390/ph14060501

**Published:** 2021-05-24

**Authors:** Joon Mo Kim, Sang Woo Park, Mincheol Seong, Seung Joo Ha, Ji Woong Lee, Seungsoo Rho, Chong Eun Lee, Kyoung Nam Kim, Tae-Woo Kim, Kyung Rim Sung, Chan Yun Kim

**Affiliations:** 1Department of Ophthalmology, Kangbuk Samsung Hospital, Sungkyunkwan University School of Medicine, 29 Saemunan-ro, Jongno-gu, Seoul 03181, Korea; kjoonmo1@gmail.com; 2Department of Ophthalmology, Chonnam National University Medical School and Hospital, 42 Jebong-ro, Dong-gu, Gwangju 61469, Korea; exo70@naver.com; 3Department of Ophthalmology, Hanyang University Guri Hospital, Kyougchun-ro 153, Guri-si 11923, Korea; goddns76@hanmail.net; 4Department of Ophthalmology, Soonchunhyang University Seoul Hospital, Soonchunhyang University College of Medicine, Seoul 04401, Korea; sjha@schmc.ac.kr; 5Department of Ophthalmology, Pusan National University School of Medicine, Busan 49241, Korea; glaucoma@pusan.ac.kr; 6Department of Ophthalmology, CHA Bundang Medical Center, CHA University, Seongnam 13496, Korea; harryrho@gmail.com; 7Department of Ophthalmology, Keimyung University School of Medicine, Daegu 42601, Korea; celee@kmu.ac.kr; 8Department of Ophthalmology, Chungnam National University College of Medicine, Daejeon 35015, Korea; kknace@cnuh.co.kr; 9Department of Ophthalmology, Seoul National University Bundang Hospital, Seoul National University College of Medicine, Seongnam 13620, Korea; twkim7@snu.ac.kr; 10Department of Ophthalmology, Asan Medical Center, University of Ulsan College of Medicine, Seoul 05505, Korea; sungeye@gmail.com; 11Department of Ophthalmology, Yonsei University College of Medicine, 50-1 Yonsei-ro, Seodaemun-gu, Seoul 03722, Korea

**Keywords:** ocular surface discomfort, dry eye, glaucoma, prostaglandin analogue, latanoprost, tafluprost, intraocular pressure

## Abstract

In this study, we investigated the effect of preservative-free (PF) 0.0015% tafluprost (TA), to the preservative containing (PC) and the PF 0.005% latanoprost (LA) in Korean subjects. This study was conducted as a multi-center, randomized, investigator-blind, active controlled, parallel-group, clinical trial in adult patients (≥19 years) with open-angle glaucoma (OAG) and ocular hypertension (OHT). After a washout period, patients with an IOP between 15 and 35 mmHg were enrolled and evaluated the efficacy, safety, and compliance at 4, 8 and 12 weeks after the first administration. A total of 137 OAG and OHT patients were randomized. Statistically significant reductions in IOP were observed in all groups. Twelve weeks after each eye drop instillation, the mean IOP reduction was −4.59 ± 2.70 mmHg (−24.57 ± 13.49%) in the PC-LA group, −4.52 ± 2.17 mmHg (−24.41 ± 11.38%) in the PF-LA, and −3.14 ± 2.83 mmHg (−17.22 ± 14.57%) in the PF-TA group. The PF-LA showed significantly better responsiveness than did PF-TA. PF-LA was better tolerated than was PC-LA. There were no adverse events that led to cessation of eye drop use in any of the groups. In conclusion, IOP decreased similarly across the groups. PF-LA may provide a good choice for OAG patients with ocular surface diseases.

## 1. Introduction

Glaucoma is a chronic progressive optic neuropathy that can cause characteristic visual field defects resulting in irreversible blindness. There are approximately 64 million people over 40 years of age with glaucoma worldwide, and this is expected to increase to approximately 76 million by 2020 and 1.11 billion by 2040 [1]. Intraocular pressure (IOP) is the most important risk factor for glaucoma, and IOP reduction is the only proven treatment, although other treatments are being proposed [2,3,4,5,6,7].

Prostaglandin analogues (PGA) are frequently used as the primary drugs for glaucoma, because they only require once daily use and effectively lower IOP with fewer systemic adverse effects than other topical glaucoma agents [2,8]. IOP is decreased by increasing the uveoscleral outflow facility [9]. However, some ocular side effects may occur, such as conjunctival injection, skin pigmentation around the eyes and iris, lengthening and thickening of the eyebrows, cystic macular edema, and recurrence of herpes keratitis [10]. Furthermore, the commonly used formulation of latanoprost (LA) eye drops contains benzalkonium chloride (BAK) and sodium phosphate, which cause frequent adverse effects [11,12]. Glaucoma patients have to use medicine chronically; therefore, it is very important to consider the adverse ocular events associated with their medications and long-term drug compliance [11]. Preservative-free (PF)-PGAs were introduced to overcome these difficulties [13,14].

Recently, a preservative-free (PF) LA eye drop (Xalost S^®^, Taejoon Pharmaceutical, Seoul, Korea) was also developed. This PF-LA contains polyoxyl 40 hydrogenated castor oil, carbomer (mucoadhesive polymer), and a high-concentration of sorbitol, which promotes substance stabilization and penetration into the eyeball (instead of BAK and sodium phosphate). In order to improve tolerability, PF-LA has a physiologically active pH range of 7.0–7.3, instead of a pH of 5.5 like that of conventional LA eye drops. This study sought to compare PF-LA with preservative containing (PC)-LA and PF-tafluprost (TA) for IOP-lowering efficacy, corneal status improvement efficacy, safety, and tolerability in patients with open-angle glaucoma (OAG) and ocular hypertension (OHT).

## 2. Results

Informed consent was obtained from all 148 patients. Of these, 137 patients were randomized and prescribed the medication. Eleven patients dropped out, while 126 subjects completed the final follow-up of this study. There were 131 subjects in the full analysis set (FAS) after excluding 6 who withdrew consent and 122 subjects in the per protocol set (PPS) who completed all schedules without protocol deviation. The FAS includes the randomized subjects who took the medication at least once, and for whom efficacy evaluations were performed at least once within a period of 12 weeks from baseline after medication administration. For a missing value, the last observation was carried forward. The PPS (per-protocol set) included those in the FAS that completed the clinical trial according to the protocol.

There were no statistically significant differences between the background patient characteristics with respect to gender, age, and duration of glaucoma regardless of group (Table 1).

### 2.1. Efficacy Evaluation

Figure 1 shows the IOP change from baseline to weeks 4, 8 and 12 in the full analysis set after eye drop instillation. The average IOP were as follows: 14.09 ± 2.87 mmHg at 4 weeks, 13.79 ± 2.58 mmHg at 8 weeks, and 13.71 ± 2.81 mmHg at 12 weeks in the PC-LA group; 14.21 ± 2.97 mmHg at 4 weeks, 13.76 ± 2.41 mmHg at 8 weeks, and 14.01 ± 2.86 mmHg at 12 weeks in the PF-LA group; and 14.24 ± 2.49 mmHg at 4 weeks, 14.38 ± 2.82 mmHg at 8 weeks, and 14.53 ± 2.71 mmHg at 12 weeks in the PF-TA group. The decreased IOP was maintained in all groups at all visits similarly, except at 8 weeks. There was a significantly higher IOP with PF-TA than there was with PF-LA at 8 week (*p* = 0.0326).

To verify the responsiveness and efficacy of each drug, we divided all subjects into responders and non-responders based on a 10% IOP decrease at week 12. The numbers of responders in the PP set were 37/42 in PF-LA, 33/40 in PC-LA, and 28/40 in PF-TA. These in the FAS set were 40/45 in PF-LA, 35/43 in PC-LA, and 29/43 in PF-TA. The PF-LA group showed significantly more responders than did PF-TA (*p* = 0.0433 in PPS, *p* = 0.0145 in FAS). However, there were no statistically difference between the PF-LA and PC-LA groups. Table 2 shows a statistically significant decreased in IOP compared with baseline in each group at week 12 (*p* < 0.0001). The IOP decreased significantly after instillation of the eye drops in all groups. The degree to which the IOP decreased was not significantly different between the groups in responders.

### 2.2. Safety Evaluation for Ocular Surface

#### 2.2.1. Corneal Staining Scores

Table 3 shows the changes in fluorescein corneal staining scores from baseline to all follow-up points, and changes in the corneal staining score between groups. All of the fluorescein corneal staining scores in group PF-LA decreased significantly from baseline at all visits. At the first visit after instillation (4 weeks), only the PF-LA group showed a decreased corneal staining score.

#### 2.2.2. Change in Hyperemia Scores (Bulbar)

Table 4 shows a change in the conjunctival congestion score with Efron Grading Scales. The hyperemia score (Bulbar) increased in PF-LA at week 4 and week 12 (*p* = 0.0049; PPS week 4, *p* = 0.0013; FAS week 4, *p* = 0.0437; FAS week 12). There were no other significant statistically changes in the hyperemia score at any follow-up points.

#### 2.2.3. Changes in Tear Break-Up Time (BUT) Scores

PF-LA led to the most marked improvement in the BUT at week 4 from baseline compared with PC-LA in both the PPS and FAS (*p* = 0.0234; PPS week 4, *p* = 0.0183; FAS week 4). However, there was no significant difference in the BUT among the three groups at any other time points (Table 5).

#### 2.2.4. Changes in OSDI (Ocular Surface Diseases Index)

Table 6 showed that ‘stinging/burning’ symptom scores in PF-LA compare to PC-LA were significantly decreased. (*p* = 0.0001; weeks 4, *p* = 0.0044; weeks 8, *p* <0.0001; weeks 12). PF-LA use led to significantly improved ‘itching’ at week 12 compared to PF-TA (*p* = 0.0336). PF-LA also led to statistically improved ‘dryness’ at week 4 compared to PF-TA (*p* = 0.0443). PF-LA also led to significantly improved ‘light sensitivity’ at weeks 4 (*p* = 0.0341) and 12 (*p* = 0.0327) compared to PC-LA. PF-LA use significantly improved ‘pain or soreness’ at week 4 compared to PC-LA (*p* = 0.0048). ‘Pain or soreness’ improved significantly with PF-TA (*p* = 0.0311) and PF-LA at week 12 compared to PC-LA (*p* = 0.0001). None of the following symptoms were considered clinically meaningful: ‘sticky eye sensation’, ‘blurred vision’, and ‘sandiness/grittiness’.

### 2.3. Adverse Events

A totally of 71 adverse events developed in 49 of 131 subjects who received investigational drugs at least once after excluding 6 withdrawn subjects. Thirty-one ocular adverse events developed in 23 subjects.

There were two cases each of ‘dry eye’ and ‘ocular discomfort’ and 1 case each of ‘eye pain’, ‘ocular hyperemia’, ‘conjunctival irritation’, and ‘conjunctival hyperemia’ in the PF-LA group. In the PC-LA group, there were three cases of ‘ocular hyperemia’; two cases each of ‘eye allergy’ and ‘eye irritation’; and one case each of ‘eyelid swelling’, ‘photophobia’, and ‘ocular discomfort’. In the PF-TA group, there were four cases of ‘eye pruritus’, three cases of ‘conjunctival hyperemia’, and one case each of ‘dry eye’, ‘eye irritation’, ‘allergic conjunctivitis’, and ‘ocular hyperemia’. The moderate adverse drug reactions were one case each of ‘dry eye’ and ‘ocular hyperemia’ in the PF-LA group and one case of ‘conjunctival hyperemia’ in PF-TA group; all other adverse drug reactions were mild. There were no severe ocular adverse drug reactions. One case of ‘conjunctival hemorrhage’ in the PF-TA group and one case of ‘hordeolum’ in the PF-LA group were not drug-related.

Non-ocular adverse events were two cases each of ‘nasopharyngitis’ and ‘bronchitis’ and one case of ‘headache’ in the PF-LA group. In the PC-LA group, there were three cases of ‘nasopharyngitis’ and two cases each of ‘headache’ and ‘bronchitis’. In the PF-TA group, there were one case each of ‘bronchitis’ and ‘headache’. There was no significant difference in incidence of ocular and non-ocular adverse events in the PF-LA group compared to those in the PC-LA group and the PF-TA group (*p*-value > 0.05).

Systemic adverse events not drug-related were irritable bowel syndrome, nasal discomfort, angina pectoris, abdominal pain, dyspepsia, ileus paralytic, dizziness postural, paresthesia, gallbladder polyp, and hepatic steatosis in the PF-LA group; facial paralysis, sudden hearing loss, colitis, gastroesophageal reflux disease, cervicitis (PC-LA), ankle fracture (PC-LA), ligament sprain (PC-LA), bursitis (PC-LA), musculoskeletal pain, cognitive disorder, and hand dermatitis in the PC-LA group; and insomnia, chest pain, non-cardiac chest pain, Irritable bowel syndrome, and gastroenteritis in the PF-TA group.

## 3. Discussion

In our study, all PG analogues led to a significant reduction in the IOP after treatment. In particular, the PF-LA group appeared to have better IOP lowering ability than did PF-TA. However, when responders were analyzed separately, there was no difference in the IOP changes between the groups. Therefore, we believe that this result was caused by the fact that there were significantly more responders in the PF-LA group than in the PF-TA group. PF-LA was significantly better tolerated than was PC-LA. This difference is probably because PF-LA does not contain BAK. However, despite the absence of BAK, PF-TA did not show significantly better tolerability than PC-LA. Additional aspects to be considered are that PF-LA contains polyoxyl 40 hydrogenated castor oil, carbomer (mucoadhesive polymer), and a high-concentration of sorbitol, all of which are used to promote substance stabilization and penetration into the eyeball (instead of BAK and sodium phosphate). It is possible that these differences in composition of the eye drops influenced their tolerability.

IOP is a crucial factor that is associated with the development and progression of glaucoma. PGA mainly reduces IOP by increasing uveoscleral outflow. The main theory is that reconstruction of the extracellular matrix increases the uveoscleral outflow of the aqueous humor. When PGA activates prostaglandin F2a, matrix metalloproteins (MMPs) are expressed in the ciliary muscle and decompose many types of collagen. This decomposition results in dilation of the ciliary muscle tissue, which increases uveoscleral outflow of aqueous humor and reduces the IOP [9,15]. Rouland et al. found that PC-LA had a similar IOP lowering effect to that of PF-LA; however, PF-LA had a lower incidence of conjunctival hyperemia than did PC-LA and improved subjective eye symptoms [16]. While, Aptel et al. reported that PC-LA and PF-LA had similar hypotensive effect and no difference in tolerance [17]. Tokuda et al. reported no significant difference in IOP after switching to PF-TA or PC-TA from PC-LA [18]. Uusitalo et al. reported that changing from PC-LA to a PF-TA formulation, PF-TA exhibited a similar IOP lowering effect and better tolerance than did PC-LA [19]. In our study, the average of IOP at 12 weeks decreased from baseline in all groups, as follows: 4.59 ± 2.70 mmHg (−24.57 ± 13.49%) in the PC-LA group; −4.52 ± 2.17 mmHg (−24.41 ± 11.38%) in the PF-LA group; and −3.14 ± 2.83 mmHg (−17.22 ± 14.57%) in the PF-TA group. In this study, PF-LA was better tolerated than PC-LA. In contrast, there was no significant difference between the tolerability of PF-LA and PF-TA.

In terms of corneal staining score, decreased fluorescein scores represent an improvement in dry eye. All of the fluorescein corneal staining scores in the PF-LA group decreased significantly from baseline at all visits. There was a statistically significant improvement at 4 weeks after instillation (vs PC-LA and PF-TA, *p* < 0.05), and this effect persisted after 4 weeks. In contrast, PF-TA did not improve 4 weeks after instillation, but showed improvement afterwards. With PC-LA, the score increased after instillation, leading to deterioration (and indicating a worsening corneal condition). This deterioration did not improve until week 12. In terms of OSDI, stinging/burning and pain/soreness were significantly lower in the PF-LA group than they were in the PC-LA group. At an early stage (4 week after instillation), the PF-LA group showed a significantly low incidence of dryness and light sensitivity than did the PF-TA group. In several studies, corneal endothelial cell damage was reportedly caused by the use of anti-glaucoma eye drops. BAK, which is used in most anti-glaucoma eye drops, can cause corneal endothelial damage and inflammation on the eye surface [17,20,21,22]. Kown J et al. suggested that this preservative was the main cause of corneal endothelial toxicity in a comparative study of PC and PF dorzolamide/timolol fixed combination eye drops [23]. Tokuda et al. reported improved superficial punctate keratopathy after switching from PC-LA to PF-TA [18].

Carbomer is an anionic polymer that strongly interacts with anionic mucin, which allows it to be widely used for artificial tears [24,25,26]. This mucoadhesive interaction causes carbomer-based formulations to bind with the mucin layer to prolong adhesion that allows for the significantly longer ocular retention time of carbomer gel compared to that of other low-viscosity eye drops [27,28,29]. The properties of carbomer seem to play a role in reducing ocular adverse events. Therefore, latanoprost may stay on the surface of the eye longer, possibly resulting in a better IOP reduction and improved corneal staining score. However, these properties may also explain why the conjunctival injection score was relatively high with latanoprost. The hyperemia score (Bulbar) appeared to increase with PF-LA use at weeks 4 and 12. The longer retention of latanoprost in the conjunctival sac in PF-LA than PC-LA due to greater hyperemia caused by the carbomer. Generally, a change of >0.7 points in the Efron scale is considered clinically meaningful. In this study, none of the changes was >0.7 points. Therefore, the increased hyperemia scores of PF-LA at weeks 4 and 12 were not clinically meaningful. The conventional latanoprost formulation (PC-LA) has a low pH (5.5). In contrast, PF-LA has a physiologically active pH range of 7.0–7.3. These improvements may lead to better tolerability. The optimal pH range to prevent corneal damage was known as 6.5 to 8.5 and the pH of lacrimal fluid is approximately pH 7.4 [30]. It is possible that this non-physiologic pH also affected tolerance.

El Hajj Moussa et al. [31] described side effects that occur in patients treated with 0.005% LA or 0.0015% TA, which included keratitis (71.4% and 100%, respectively), conjunctivitis (57.1% and 33.3%), and conjunctival hyperemia (55.1% and 55%). The LA group showed additional side effects of lengthening and curling of eyelashes, pigmentation of the iris, and recurrence of herpes keratitis. We did not evaluate all of these adverse effects from PGA in this study (such as lid pigmentation, deepening of the upper eyelid sulcus, and eyebrow growth) given the short follow-up duration.

Our study had several limitations such as the relatively small number of subjects. In addition, this study was performed using data from one ethnic group. Therefore, our results may not be applicable to other ethnic groups. Our study was multi-center, randomized, and investigator-blind to reduce bias and generalize our results. However, since few of a large number of invited institutions participated, it is possible that the variability of the tests among institutions had an influence on our results. To reduce this, training and monitoring were conducted at all institutions, though this cannot eliminate all bias. Moreover, we conducted the study at intervals of 4 weeks to allow washout (+1 week for the window period). This follows the generally accepted washout period of latanoprost [17]. However, since drug responses can vary by patient, we cannot guarantee that this washout period was appropriate. The baseline IOP of this study was lower than that of another study, which could affect the efficacy and responsiveness. To compare our result with that of another study, our baseline IOP should be considered. In addition, we did not evaluate all adverse effects of the prostaglandin analogues due to the relatively short follow-up duration. Finally, we did not measure the 24-h IOP variation. However, despite the above limitations, our data provide comparative information between PF-LA, PC-LA, and PF-TA in terms of IOP reduction and ocular surface adverse effects. We believe that these data will serve as a good clinical reference.

## 4. Materials and Methods

This study was conducted as a multi-center, randomized, investigator-blind, active controlled, parallel-group, clinical trial in adult patients (≥19 years) with OAG and OHT. It was conducted in 15 clinical sites between December 19, 2018 and December 9, 2019. The study protocol was approved by the Institutional Review Board (IRB) at each institution and performed in accordance with the Declaration of Helsinki. The principal investigator’s representative IRB registration number is 4-2018-0929 and its IRB approval date is 22 November 2018. ClinicalTrials.Gov Identifier of this study is NCT04164459.

The primary objective was to demonstrate the superiority of the trial drug (PF-LA) to the control drug (PC-LA) in terms of corneal staining score (oxford grade) variation after administration of the drugs for 12 weeks. Anastasis-Georgoios et al. referenced a superiority difference value of 0.9 [32]. The other main objective was to demonstrate the non-inferiority of the trial drug (PF-LA) to the control drug (PF-TA) in terms of corneal staining score (oxford grade) variation after administration of the drugs for 12 weeks. The upper limit of non-inferiority was set at 0.45 as the standard acceptance level in glaucoma studies. The adjusted average and standard error of the corneal staining score variations in the trial and control groups, the difference between the average and adjusted average, 95% two-tailed confidence interval of adjusted average difference, and *p*-value were calculated by analysis of covariation (ANCOVA) with baseline IOP as the covariate.

All patients underwent complete baseline ophthalmic examinations for regular glaucoma monitoring. These examinations included: slit-lamp examination, gonioscopic examination, best-corrected visual acuity (BCVA), fundus photography, IOP measurement using a Goldmann applanation tonometer (Haag-Streit, Kornig, Switzerland), visual field (VF) test (Humphrey visual field analyzer, Carl Zeiss Meditec, Dublin, CA, USA), and ultrasonic corneal pachymetry. Personal medical histories were also recorded, including prior ocular disease and surgeries, systemic diseases (including diabetes and hypertension), as well as other medications. Patients were excluded if they had corneal abnormalities or ocular diseases other than glaucoma, underwent ocular surgery other than cataract surgery, did not receive regular ophthalmic examinations, or did not apply the prescribed eye drops. IOP was measured continuously three times in the same eye using a Goldmann applanation tonometer by an ophthalmologist, and the average value was recorded. In enrolled subjects after the washout period, IOP was measured at 10 a.m. (±1 h) in both eyes at every visit. IOP was measured in the right eye first and then in the left eye.

This study enrolled adult patients (≥19 years of age) with OAG/OHT. Glaucoma was defined based on characteristics of the optic disc, such as presence of diffuse or localized rim thinning, rim notching, and/or retinal nerve fiber layer (RNFL) defect with glaucomatous VF defect. Typical glaucomatous VF defects were based on Anderson’s criteria: (1) Glaucoma Hemifield Test (GHT) result “outside normal limits”; (2) three contiguous non-edge points on the pattern deviation plot within the Bjerrum’s area with *p* < 0.05, one of which is *p* < 0.01; and (3) pattern standard deviation (PSD) with *p* < 0.05 [33]. Eligible eyes with an IOP between 15 mmHg and 35 mmHg at 10 a.m. (±1 h) after a washout period were randomized and assigned to a treatment schedule with daily PF-LA (Xalost S^®^, Taejoon Pharmaceutical Co., Ltd., Yongin, Korea), PC-LA (Xalatan^®^, Pfizer Manufacturing Belgium NV/SA, Puurs, Belgium), or PF-TA (Taflotan-S^®^, Santen, Osaka, Japan). We excluded patients with 20/80 or lower best-corrected visual acuity on the Snellen chart, and those with a medical history of chronic intraocular inflammation within 3 months of screening. We also excluded patients who used contact lenses during the clinical study, and those who were pregnant, planning to become pregnant, nursing, or of childbearing potential and not using a reliable form of contraception. If both eyes met the criteria, the eye with the higher IOP was selected. If the IOP was equal in both eyes, then the right eye was selected. After the washout period for 4 weeks [34], the subjects were randomized 1:1:1 to monotherapy with PF-LA, PC-LA, or PC-TA. A total of 137 patients were randomized to each eye ophthalmic solution group. The patients were instructed to instill one drop in each eye once daily in the evening (9 PM ± 1 h). All patients were scheduled for follow-up visits at 4, 8, and 12 weeks. Figure 2 shows the flow chart of this study.

At each follow-up visit, the following evaluations were conducted consecutively: tear break-up time (BUT), corneal staining, conjunctival staining, congestion score, IOP measurement, best-corrected visual acuity change, and a questionnaire evaluating tolerability with the Ocular Surface Disease Index (OSDI). The conjunctival hyperemia scores were evaluated at each visit (and compared to the baseline before fluorescein staining). Using the Efron Grading Scales, each evaluation was performed on bulbar and limbal conjunctiva [35]. After applying fluorescein with fluorescein paper stick, the tear BUT was evaluated under slit lamp illumination of a cobalt blue light source. We observed the point where black spots, streaks, or fluorescein defects occurred in the tear layer stained with fluorescein after the patient blinked 2–3 times [36]. The time was measured in seconds. The measurement result was repeated three times, and the average value was used. Corneal staining was evaluated according to the Oxford grading system with fluorescein staining [37]. Conjunctival staining was evaluated according to the National Eye Institute (NEI) scale with fluorescein staining [38]. The OSDI questionnaire consists of 12 questions, and the total score is measured by dividing the total of each score by the number of questions (0 to 4 points for each question). It is expressed as a score from 0 to 100. The larger the score, the more severe the symptoms. According to the score distribution, the OSDI is classified into normal (0–12 points), mild (13–22 points), moderate (23–32 points), and severe (33 points or more) [39].

### Statistical Analysis

Data are shown as means ± standard errors. Monocular data analyses of the eligible eyes were performed for statistical comparisons. The efficacy analyses consisted of the corneal staining score, IOP, BUT, hyperemia score and OSDI (Ocular surface diseases index) at 4, 8, and 12 weeks. The statistical differences between baseline and each week were evaluated using the paired t-test or Wilcoxon signed rank test. Statistical differences between the treatment and experimental groups were evaluated using analysis of covariance (ANCOVA). A *p*-value < 0.05 was considered statistically significant.

## 5. Conclusions

In conclusion, our results suggest that all PGAs sufficiently reduce IOP. However, among the three PGAs, PF-LA was associated with the best patient responsiveness and tolerability.

## Figures and Tables

**Figure 1 pharmaceuticals-14-00501-f001:**
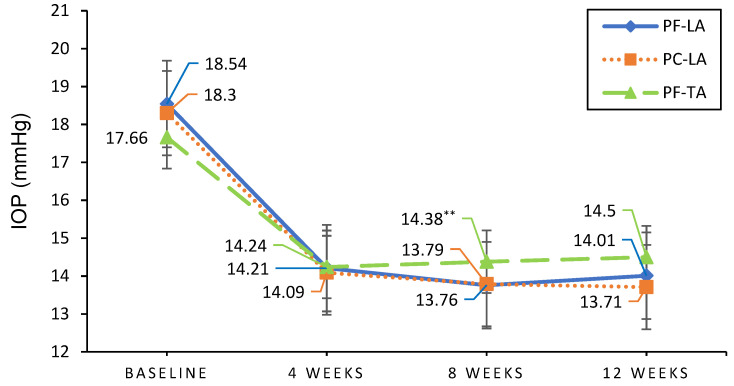
IOP changes from baseline to 4, 8 and 12 weeks in each group of the per-protocol set. PF-TA led to a significantly higher IOP compared to that of PF-TA at week 8 (*p* = 0.0326) The average IOP was as follows: 14.09 ± 2.87 mmHg at 4 weeks, 13.79 ± 2.58 mmHg at 8 weeks, and 13.71 ± 2.81 mmHg at 12 weeks in the PC-LA group; 14.21 ± 2.97 mmHg at 4 weeks, 13.76 ± 2.41 mmHg at 8 weeks, and 14.01 ± 2.86 mmHg at 12 weeks in the PF-LA group; and 14.24 ± 2.49 mmHg at 4 weeks, 14.38 ± 2.82 mmHg at 8 weeks, and 14.53 ± 2.71 mmHg at 12 weeks in the PF-TA group. ** *p* = 0.0326.

**Figure 2 pharmaceuticals-14-00501-f002:**
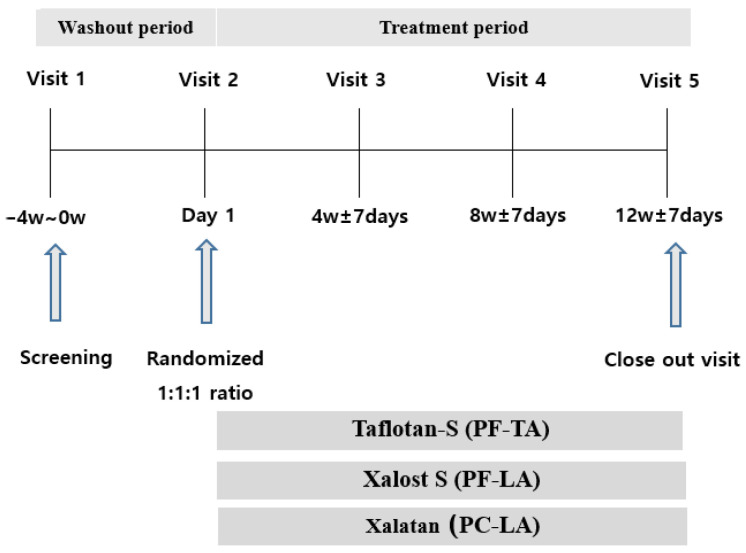
Flow chart. After the washout period, subjects were randomized 1:1:1 to monotherapy with PF-LA, PC-LA, or PC-TA. Patients were instructed to instill one drop in each eye once daily (9 PM ± 1 h) and were scheduled for follow-up visits at 4, 8, and 12 weeks.

**Table 1 pharmaceuticals-14-00501-t001:** Subject disposition and demographics. Demographic data were used for full analysis set (FAS).

Subjects	PF-LA	PC-LA	PF-TA	Total	*p*-Value
Randomized	46	46	45	137	
Completed	44	42	40	126	
Withdrawal	2	4	5	11	
Withdrawal reason					
Protocol violation	1	0	0	1	
Withdrawal of consent	1	2	3	6	
Failure to follow-up	0	1	0	1	
Advance events	0	0	1	1	
Non-compliance of inclusion/exclusion criteria	0	1	1	2	
FAS (full analysis set)	45	43	43	131	
PPS (per-protocol set)	42	40	40	122	
Gender, female, N (%) *	21 (46.67)	20 (46.51)	17 (39.53)	58 (44.27)	0.7471 ^†^
Age (years), Mean (SD)	56.69 (12.96)	57.44 (11.78)	56.81 (13.78)	56.98 (12.77)	0.9532 ^‡^
Duration of glaucoma (years (y)/months (m)), Mean (SD)	3y 9m (4y 3m)	3y 2m (4y 9m)	2y 6m (3y 6m)	3y 3m (4y 4m)	0.3231 ^‡^
Hypertension, N (%)	16 (35.56)	15 (34.88)	12 (27.91)	43 (32.82)	
Diabetes mellitus, N (%)	11 (24.44)	7 (16.28)	4 (9.30)	22 (16.79)	

* FAS. ^†^ Chi-square test. ^‡^ Kruskal–Wallis test.

**Table 2 pharmaceuticals-14-00501-t002:** IOP changes from baseline to 12 weeks (mean ± SD, mmHg) of the per-protocol set (PPS) and full analysis set (FAS) groups.

	PF-LA		PC-LA		PF-TA		*p* Value ^§^	*p* Value *	*p* Value ^†^	*p* Value ^‡^
	Baseline	12 Weeks	Baseline	12 Weeks	Baseline	12 Weeks				
PPS	18.54 ± 2.76	14.01 ± 2.86	18.30 ± 2.92	13.71 ± 2.81	17.66 ± 2.33	14.53 ± 2.71	0.3651	<0.0001	0.7527	0.0515
FAS	18.54 ± 2.77	13.92 ± 2.86	18.28 ± 2.82	13.86 ± 2.80	18.19 ± 3.53	15.19 ± 4.25	0.5700	<0.0001	0.7948	0.0043
PPS Responder	18.57 (2.89)	13.54 (2.70)	18.61 (3.01)	13.20 (2.72)	17.75 (2.57)	13.23 (1.92)	0.4591	<0.0001	0.3767	0.6230
FAS Responder	18.58 (2.89)	13.48 (2.70)	18.60 (2.93)	13.29 (2.67)	17.72 (2.53)	13.26 (1.89)	0.4199	<0.0001	0.6069	0.4335

* Significant difference in IOP change at week 4, week 8, and week 12 of administration as compared with baseline in each group (paired *t*-test); ^†^ Difference in IOP between PF-LA and PC-LA at 12 weeks (ANCOVA model); ^‡^ Difference in IOP between PF-LA and PF-TA at 12 weeks (ANCOVA model), ^§^ Difference between control and treatment group at baseline (Kruskal–Wallis test) *p*-value < 0.05.

**Table 3 pharmaceuticals-14-00501-t003:** Corneal staining score measurement time at 4, 8, and 12 weeks (mean ± SD, mmHg) of the per-protocol set (PPS) and full analysis set (FAS) groups.

		Baseline	4 Weeks	*p* Value *	*p* Value ^†^	8 Weeks	*p* Value *	*p* Value ^‡^	12 Weeks	*p* Value *	*p* Value ^§^
PPS	PF-LA	0.81 ± 0.99	0.36 ± 0.53	0.0004		0.52 ± 0.63	0.0559		0.48 ± 0.74	0.0309	
	PC-LA	0.58 ± 0.78	0.83 ± 0.96	0.1196	0.0006	0.53 ± 0.64	0.8560	0.6438	0.73 ± 0.82	0.332	0.0431
	PF-TA	0.70 ± 0.79	0.73 ± 0.88	0.8909	0.0003	0.60 ± 0.67	0.3930	0.6956	0.58 ± 0.59	0.3111	0.0666
FAS	PF-LA	0.80 ± 0.99	0.36 ± 0.53	0.0002		0.49 ± 0.63	0.0295		0.47 ± 0.73	0.0311	
	PC-LA	0.56 ± 0.77	0.81 ± 0.93	0.0960	0.0107	0.51 ± 0.63	0.8601	0.4382	0.67 ± 0.81	0.4300	0.3996
	PF-TA	0.67 ± 0.78	0.72 ± 0.85	0.7816	0.0065	0.60 ± 0.66	0.4997	0.5379	0.58 ± 0.59	0.4095	0.3132

* Significant difference in corneal staining score change at week 4, week 8, and week 12 of administration as compared with baseline in each group (Wilcoxon signed rank test); ^†^ Difference in corneal staining score between PF-LA, PC-LA, PF-TA at 4 weeks (ANCOVA model); ^‡^ Difference in corneal staining score between PF-LA, PC-LA, PF-TA at 8 weeks (ANCOVA model); ^§^ Difference in corneal staining score between PF-LA, PC-LA, PF-TA at 12 weeks (ANCOVA model); *p*-value < 0.05.

**Table 4 pharmaceuticals-14-00501-t004:** Hyperemia score measurement time at 4, 8, and 12 weeks (mean ± SD, mmHg) of the per-protocol set (PPS) and full analysis set (FAS) groups.

Analysis	Group	Baseline	4 Weeks	*p* Value *	*p* Value ^†^	8 Weeks	*p* Value *	p Value ^‡^	12 Weeks	*p* Value *	*p* Value ^§^
PPS	PF-LA	0.67 ± 0.69	0.98 ± 0.81	0.0049		0.86 ± 0.81	0.1396		0.88 ± 0.80	0.0649	
	PC-LA	0.73 ± 0.75	0.75 ± 0.67	1.0000	0.0343	0.78 ± 0.70	0.7813	0.3946	0.75 ± 0.71	1.0000	0.2022
	PF-TA	0.78 ± 0.77	0.95 ± 0.64	0.1907	0.4507	0.88 ± 0.69	0.5034	0.7479	0.80 ± 0.65	1.0000	0.2646
FAS	PF-LA	0.67 ± 0.67	1.02 ± 0.87	0.0013		0.84 ± 0.80	0.1396		0.89 ± 0.78	0.0437	
	PC-LA	0.74 ± 0.76	0.74 ± 0.66	1.0000	0.0079	0.77 ± 0.68	1.0000	0.0899	0.72 ± 0.70	1.0000	0.0899
	PF-TA	0.79 ± 0.77	0.95 ± 0.65	0.1907	0.2264	0.88 ± 0.70	0.5034	0.2433	0.81 ± 0.66	1.0000	0.2433

* Significant difference in hyperemia score (Bulbar) change at week 4, week 8, and week 12 of administration as compared with baseline in each group (Wilcoxon signed rank test); ^†^ Difference in hyperemia score (Bulbar) between PF-LA, PC-LA, and PF-LA at 4 weeks (ANCOVA model); ^‡^ Difference in hyperemia score (Bulbar) between PF-LA, PC-LA, and PF-LA at 8 weeks (ANCOVA model); ^§^ Difference in hyperemia score (Bulbar) between PF-LA, PC-LA, and PF-LA at 12 weeks (ANCOVA model).

**Table 5 pharmaceuticals-14-00501-t005:** BUT at 4, 8 and 12 weeks (mean ± SD, mmHg) of the per-protocol set (PPS) and full analysis set (FAS) groups.

Analysis	Group	Baseline	4 Weeks	*p* Value *	*p* Value ^†^	8 Weeks	*p* Value *	*p* Value ^‡^	12 Weeks	*p* Value *	*p* Value ^§^
PPS	PF-LA	6.44 ± 2.47	6.83 ± 2.53	0.3160		6.63 ± 3.39	0.7106		6.14 ± 2.45	0.8213	
	PC-LA	6.06 ± 2.73	5.55 ± 2.27	0.1464	0.0234	6.72 ± 3.01	0.3900	0.6662	6.16 ± 2.86	0.9182	0.7603
	PF-TA	6.07 ± 2.88	5.89 ± 2.67	0.7117	0.1138	5.71 ± 3.00	0.4790	0.2507	5.75 ± 2.86	0.2266	0.6726
FAS	PF-LA	6.29 ± 2.46	6.75 ± 2.49	0.2133		6.58 ± 3.27	0.5297		6.10 ± 2.40	0.9510	
	PC-LA	6.00 ± 2.68	5.53 ± 2.20	0.1575	0.0183	6.61 ± 2.94	0.4016	0.7828	6.13 ± 2.76	0.7727	0.7776
	PF-TA	6.00 ± 2.85	5.90 ± 2.68	0.8172	0.1143	5.73 ± 2.99	0.6356	0.2350	5.77 ± 2.86	0.3488	0.6829

* Significant difference in BUT change at week 4, week 8, and week 12 of administration as compared with baseline in each group (Wilcoxon signed rank test); ^†^ Difference in BUT between PF-LA, PC-LA, and PF-LA at 4 weeks (ANCOVA model); ^‡^ Difference in BUT between PF-LA, PC-LA, and PF-LA at 8 weeks (ANCOVA model); ^§^ Difference in BUT between PF-LA, PC-LA, and PF-LA at 12 weeks (ANCOVA model).

**Table 6 pharmaceuticals-14-00501-t006:** Ocular surface diseases index (OSDI) evaluated at 4, 8, and 12 weeks (mean ± SD, mmHg) in the Per-Protocol Set (PPS) group.

		PF-LA	PC-LA		PF-TA	
				*p* Value *		*p* Value ^†^
Stinging/burning	4 Weeks	0.33 ± 0.69	0.95 ± 0.81	0.0001	0.58 ± 0.75	0.0770
	8 Weeks	0.33 ± 0.69	0.74 ± 0.79	0.0044	0.50 ± 0.72	0.1507
	12 Weeks	0.17± 0.44	0.75 ± 0.71	<0.0001	0.45 ± 0.78	0.0787
Sticky eye sensation	4 Weeks	0.21 ± 0.52	0.25 ± 0.63	0.9153	0.20 ± 0.46	0.9660
	8 Weeks	0.21 ± 0.56	0.33 ± 0.58	0.1700	0.15 ± 0.43	0.7658
	12 Weeks	0.15 ± 0.36	0.35 ± 0.62	0.0924	0.20 ± 0.41	0.5304
Itching	4 Weeks	0.33 ± 0.61	0.40 ± 0.50	0.3245	0.45 ± 0.75	0.5767
	8 Weeks	0.24 ± 0.48	0.33 ± 0.58	0.4614	0.58 ± 0.90	0.1018
	12 Weeks	0.22 ± 0.47	0.45 ± 0.64	0.0691	0.55 ± 0.78	0.0336
Blurred vision	4 Weeks	0.43 ± 0.70	0.40 ± 0.63	0.9418	0.55 ± 0.81	0.5646
	8 Weeks	0.38 ± 0.73	0.56 ± 0.88	0.2697	0.70 ± 0.94	0.0945
	12 Weeks	0.34 ± 0.69	0.50 ± 0.64	0.1209	0.53 ± 0.75	0.1951
Sandiness/grittiness	4 Weeks	0.38 ± 0.66	0.43 ± 0.59	0.5650	0.53 ± 0.82	0.5267
	8 Weeks	0.36 ± 0.62	0.41 ± 0.64	0.6660	0.60 ± 0.84	0.2031
	12 Weeks	0.32 ± 0.57	0.45 ± 0.68	0.3850	0.50 ± 0.60	0.1117
Dryness	4 Weeks	0.29 ± 0.71	0.35 ± 0.70	0.4273	0.58 ± 0.81	0.0443
	8 Weeks	0.36 ± 0.58	0.38 ± 0.63	0.9488	0.40 ± 0.71	1.0000
	12 Weeks	0.29 ± 0.51	0.53 ± 0.78	0.2153	0.45 ± 0.78	0.5699
Light sensitivity	4 Weeks	0.14 ± 0.42	0.35 ± 0.62	0.0746	0.50 ± 0.88	0.0341
	8 Weeks	0.31 ± 0.72	0.31 ± 0.52	0.4739	0.43 ± 0.71	0.2291
	12 Weeks	0.17 ± 0.50	0.45 ± 0.75	0.0327	0.33 ± 0.69	0.2305
Pain or soreness	4 Weeks	0.12 ± 0.50	0.43 ± 0.68	0.0048	0.33 ± 0.62	0.0311
	8 Weeks	0.24 ± 0.48	0.36 ± 0.58	0.3279	0.28 ± 0.51	0.7137
	12 Weeks	0.02 ± 0.16	0.48 ± 0.68	0.0001	0.18 ± 0.59	0.1569

* Difference between PF-LA and PC-LA (Wilcoxon rank sum test); ^†^ Difference between PF-LA and PF-TA (Wilcoxon rank sum test).

## Data Availability

The datasets generated and/or analyzed during this study are not publicly available due to data protection laws for human research but may be available from the corresponding author on reasonable request.

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
