# Peer review of "Comparison of the Safety and Efficacy between Preserved and Preservative-Free Latanoprost and Preservative-Free Tafluprost"

_pharmaceuticals, 2021, doi:10.3390/ph14060501_

Round 1

Reviewer 1 Report

Very good article. I do not have suggestions.

Author Response

Thank you for your encouragement.

Reviewer 2 Report

It is an interesting paper worthy for publication. Would suggest only minor comments in order to clarify 2 points :

  1. A literature reference should be added to define glaucoma
  2. In Figure 1, PC-TA, PF-LA and PC-LA should be added because only the names of the specialties appear

Author Response

I have attached the answer to the file.

Reviewer 3 Report

Introduction:

- prevalence of glaucoma is much higher - about 79 mil in 2020.

Material and methods:

- complete methodology description is lacking - for instance, IOP measurements, timing

  • page 4 After applying blue fluorescein.... blue-fluorescein staining. Fluorescein staining is the standard notation

Results

  • - 126 completed the study? However, this number is different from either FAS or PPS. It is not clear what ~completed the study~ means (ussually, it means patients adherent to protocol)
  • IOP results is showed in Fig.1 for FAS. I suggest to change that to PPS, or include that, too. FAS was defined as patient who took the medication at least once, any comparison in this group is statistically challangeing. PPS evaluation at 12 weeks is provided, indeed.
  • IOP at baseline is almost 1mm lower for PF-TA than the 2 other groups. Is that statistically significant? Could it explain the lower efficiency?
  • adverse events - stipulating that 133 received investigational drug (FAS=131)
  • adverse events should be rewritten to better point out the reactions and the prevalences (especially in the last paragraph, in detail)

Discussion

  • how do you explain the difference between corneal staining and conjunctival staining for PF-LA?
  • the multicenter evaluation with such a small sample o patients means a high variability, especially regarding the staining scoring, but also IOP measuring. 136 patient in 15 centers (9 patients/center) could also make the randomisation process verry difficult to controll. This concern should be adress.
  • after washout, still the IOP remained controlled in most. More details about the gaucoma condition of the patients could be providid. Low baseline IOPs could influence the eficacy of the treatment

Bibliography

  • is outdated - apart from 2-3 titles (2018, 2019), all the citations date before 2015, many before 2000
  • similar research was performed - for instance DOI: 10.1155/2017/3540749, in 2017, that should be discussed

Author Response

I have attached the answer to the file.

Reviewer 4 Report

The authors have nice data regarding differential effects of preservative and preservative free PG analogs in a moderate size sample of Korean subjects.  The paper in general is well written with some specific areas of the article requiring revision.  Specific comments are below:

  1. The comments in the introduction about IOP reduction being the “most effective” treatment for glaucoma are a bit off base in terminology… IOP reduction is the only currently approved treatment so it is unclear what other treatments might be inferred by the authors current statements. 
  2. In addition, the above statement is attributed to an NTG study from 19 years ago and the authors should use a relevant current citation when referencing the current state of treatment as well as risk factors.
  3. The term intraocular pressure is written with abbreviation early in the introduction then the abbreviation IOP is used, but then soon after the full term is written out again. This error and similar ones should be checked throughout the paper.
  4. The authors should provide a statistical power estimate for their study n and/or other justification for the sample size chosen.
  5. The use of 7 different tables and multiple figures for this data seems slitghly excessive, I would encourage the authors to eliminate 1 table if possible and avoid redundancy of written results with tabled results.

Author Response

I have attached the answer to the file.

Round 2

Reviewer 3 Report

Thank you for your response.

The only thing to adress is still the introduction.

Please change: "There are approximately 640,000 peo-ple over 40 years of age with glaucoma worldwide, and this is expected to increase to approximately 0.76 million by 2020 and 1.11 million by 2040"

The number of glaucomatous patients should be in milions (not 790.000, but 79.000.000, or 79 milions etc. The same with 640.000), according to Reference 1 and WHO. Prevalence is very high:  The global prevalence of glaucoma for population aged 40-80 years is 3.54% according to your reference (1)

Change paragraph into:

There are approximately 64 million people over 40 years of age with glaucoma worldwide, and this is expected to increase to approximately 76 million by 2020 and 1.11 billion by 2040

Author Response

The answer to the reviewer is attached as a file.
